# Fourth Dose of mRNA COVID-19 Vaccine Transiently Reactivates Spike-Specific Immunological Memory in People Living with HIV (PLWH)

**DOI:** 10.3390/biomedicines10123261

**Published:** 2022-12-15

**Authors:** Giulia Lamacchia, Lorenzo Salvati, Seble Tekle Kiros, Alessio Mazzoni, Anna Vanni, Manuela Capone, Alberto Carnasciali, Parham Farahvachi, Filippo Lagi, Nicoletta Di Lauria, Arianna Rocca, Maria Grazia Colao, Francesco Liotta, Lorenzo Cosmi, Gian Maria Rossolini, Alessandro Bartoloni, Laura Maggi, Francesco Annunziato

**Affiliations:** 1Department of Experimental and Clinical Medicine, University of Florence, 50121 Florence, Italy; 2Infectious and Tropical Diseases Unit, Careggi University Hospital, 50134 Florence, Italy; 3Flow Cytometry Diagnostic Center and Immunotherapy, Careggi University Hospital, 50134 Florence, Italy; 4Microbiology and Virology Unit, Careggi University Hospital, 50134 Florence, Italy; 5Immunology and Cell Therapy Unit, Careggi University Hospital, 50134 Florence, Italy; 6Immunoallergology Unit, Careggi University Hospital, 50134 Florence, Italy

**Keywords:** SARS-CoV-2, mRNA vaccine, fourth dose, HIV, people living with HIV, humoral response, T cell response, B cell response, Omicron

## Abstract

**Background**: People Living With HIV (PLWH), with advanced disease, lower CD4+ T cell counts or an unsuppressed HIV viral load can have a suboptimal vaccine response. For this reason, in the current COVID-19 pandemic, they represent a prioritized population for the SARS-CoV-2 fourth (or second booster) vaccine dose. This work aims to investigate the effects of a second booster on the reactivation of the spike-specific humoral and cell-mediated immune responses in PLWH. **Methods**: A total of eight PLWH, who received a fourth dose of the original mRNA vaccines were enrolled. They were evaluated before and then 7 days, 1 month and 2 months after the injection. The humoral response was assessed via a chemiluminescent immunoassay. Immunophenotyping and the functional evaluation of the SARS-CoV-2-specific cellular immune responses were performed via flow cytometry. **Results**: Anti-spike IgG levels were above the cut-off value for all subjects at all timepoints. The spike-specific CD4+ T cell response was reactivated one week after the fourth vaccine dose, and on average declined at two months post-vaccination. A similar trend was observed for the spike-specific B cells. A low percentage of spike-specific CD4+ T cells was activated by the B.1.1.529 BA.1 Omicron-spike mutated peptides, and the majority of these cells were reactive to the conserved portions of the spike protein. Similarly, the majority of the spike-specific memory B cells were able to bind both Wuhan and Omicron-spike entire protein. **Conclusions**: Spike-specific adaptive immune responses are transiently reactivated in PLWH following the fourth mRNA vaccine dose. The breadth of the immune responses to the mutated spike protein provides insight on the possible cross-reactivity for the SARS-CoV-2 variants of concern (VOCs).

## 1. Introduction

The COVID-19 vaccination has dramatically changed the course of the pandemic caused by SARS-CoV-2 [1]. As a matter of fact, the establishment of massive herd immunity benefits both not-at-risk individuals and immunocompromised individuals. As the latter are considered high-risk subjects for severe COVID-19, nowadays immunocompromised individuals are being administered with additional booster doses of eligible vaccines. People living with HIV (PLWH), with advanced HIV disease, lower CD4+ T cell counts or an unsuppressed HIV viral load are considered at high-risk, as the main hosts of HIV are CD4+ T lymphocytes, which orchestrate adaptive immunity responses and the immunological memory [2,3]. HIV is a retrovirus whose infection gradually deteriorates the functionalities of the immune response, as it induces a slow decrease of the CD4+ T cell pool and causes major gaps in the immune repertoire [4]. The end stage of HIV infection is the development of the acquired immune deficiency syndrome (AIDS), that exposes the individual to severe complications [4,5]. Data on the effective risk of a worst outcome in PLWH infected by SARS-CoV-2 is controversial [6,7,8,9]. Several studies have concluded that HIV infection is to be considered a generic co-morbidity. In contrast, others, such as the South African study [10] and the New York study [11], show that HIV-positive subjects developing COVID-19 are more likely to be hospitalized and admitted to intensive care units. At the early stages of the pandemic, a few studies reported higher values of inflammatory markers during hospital admission, in HIV+ COVID-19 patients, compared to HIV- counterparts [12]. As for the immune response of PLWH after developing COVID-19, some research shows the mounting of a cellular and humoral response with the exception of lower rates of the neutralizing antibody production, with respect to individuals without HIV [13]. Moreover, past studies observed lower responses to other vaccines, such as those against the influenza virus or pneumococcus in HIV positive individuals, most likely due to the impairment of the CD4+ T cell pool and, subsequently, the altered B cell activation and functionality [14]. However, all vaccinations can be recommended in PLWH, with the exception of live attenuated vaccines which are contraindicated if the CD4+ T cell counts are lower than 200 cells/mm^3^, as they can potentially cause infection in immunocompromised individuals [15]. Moreover, it is known that vaccinations (both T cell-dependent and T cell-independent vaccines) can induce increases in the HIV viral load even during antiretroviral therapy, thereby exposing the latent viral reservoir [16,17,18,19]. Similarly, the COVID-19 vaccination can cause transient increases in HIV viremia [20,21].

Data on the COVID-19 vaccine efficacy and immunogenicity remain scarce in PLWH [22]. It has been shown that a third mRNA vaccine dose strongly boosts the humoral immune response in PLWH at higher levels, compared to the post-second dose, regardless of the CD4+ T cell count [23]. As of October 2022, there are few reports on the efficacy of a fourth dose of the COVID-19 mRNA vaccine on the reactivation and maintenance of the immunological memory against the SARS-CoV-2-spike protein in immunocompromised patients [24,25,26]. Starting from March 2022, some countries implemented a fourth mRNA vaccine dose for the highest risk populations, including people with a moderate to severely advanced HIV disease, generally 3–5 months after the first booster dose [6]. It has been demonstrated that a fourth dose of the original BNT162b2 mRNA vaccine is effective in reducing the short-term risk of COVID-19–related outcomes, among people over 60 years of age who received a third dose at least 4 months earlier [27]. Even during the Omicron surge, a fourth vaccine dose appears to reduce the short-term risk of death from all causes among the elderly, compared to the third dose [26]. However, it has been recently shown that in mainly healthy individuals, the additional immunologic benefit of the fourth original BNT162b2 mRNA vaccine dose, in terms of infection rate, as compared with the third dose, is lower and wanes by 13 weeks following vaccination [28]. 

In this study, our purpose is to define the quality and longevity of the SARS-CoV-2-specific immunological memory induced by the fourth dose of the original mRNA vaccines in a cohort of PLWH and followed up to 2 months post-vaccination.

## 2. Materials and Methods

Study Participants: A total of eight people living with HIV (PLWH), who received a fourth dose of the original BNT162b2 mRNA vaccine, were recruited by the Infectious and Tropical Diseases Unit of Careggi University Hospital, Firenze, Italy. Between the end of March 2022 and the end of June 2022, four peripheral blood draws were collected from all the study participants: before the administration of the fourth vaccine dose (basal time), and then 7 days, 1 month and 2 months after the injection. The mean age of the cohort was 58 years old. The main demographic and clinical characteristics of the cohort are summarized in Table 1. All included subjects received a fourth mRNA vaccine dose at the end of March 2022. A negative history for infection before the first blood draw was based on the absence of positive swabs, of symptoms and of anti-nucleoprotein IgGs at the baseline (pre-fourth vaccine dose), as shown in Table 2.

Evaluation of the SARS-CoV-2-specific IgMs and IgGs: Evaluation of the SARS-CoV-2 spike protein antibodies, including the anti-spike-specific (in trimeric form) IgGs (Diasorin, Vercelli, Italy), anti-spike RBD-specific IgGs (Abbott, Rome, Italy), anti-spike-specific IgMs (Abbott, Rome, Italy), anti-nucleoprotein-specific IgGs (Abbott, Rome Italy), and neutralizing antibodies that block the binding of spike protein with the ACE2 receptor (Dia.Pro Diagnostic Bioprobes, Milan, Italy), was performed following the manufacturers’ instructions. The antibody reactivity of each specimen was expressed in BAU/mL or by the ratio between the optical density and the cut-off value (index).

Evaluation of the T cell subsets and the SARS-CoV-2 spike-reactive T cells: The PBMCs were obtained following the density gradient centrifugation of the blood samples using Lymphoprep (Axis Shield Poc As, Dundee, Scotland) and were frozen in FBS plus 10% DMSO for storage in liquid nitrogen. For each individual, the longitudinal samples were defrosted and analyzed together for the B cell and T cell evaluation. 

The T cell subset immunophenotyping was performed to obtain the CD4/CD8 ratio on the defrosted PBMCs, isolated at the pre-fourth dose timepoint. The cells were incubated with the fluorochrome-conjugated antibodies, listed in Appendix A, for 15 min at room temperature. Then, the cells were washed in a PBS solution at 0.5% BSA. The samples were acquired on a BD LSR II flow cytometer (BD Biosciences, Franklin Lakes, NJ, USA) with FACSDiva Software and analyzed by FlowJo v10 Software. The gating strategy used is presented in Appendix A.

For the T cell stimulation in vitro, 1.5 million PBMCs were cultured in complete RPMI plus 5% human AB serum in 96-well flat-bottom plates in the presence of the medium alone (background, negative control), of the superantigen staphylococcal enterotoxyn B (SEB, positive control) and of a pool of the spike SARS-CoV-2 peptide pools (Prot_S1, Prot_S+, and Prot_S to achieve a complete sequence coverage of the spike protein) at 0.6 μM/peptide, accordingly to the manufacturer’s instructions (Miltenyi Biotech, Bergish Gladbach, Germany). Prot-S1 covers the complete N-terminal S1 domain (aa 1–692), Prot_S+ parts of the C-terminal S2 domain (aa 689–895), while the Prot_S peptide pools cover parts of the S1 and S2 domains (aa 304–338, 421–475, 492–519, 683–707, 741–770, 785–802, 885–1273). Combining the Prot_S+ and Prot_S peptide pools, the complete S2 domain is covered. Alternatively, the cells were stimulated with peptide pools selectively spanning the mutated regions in the Omicron (B.1.1.529) BA.1 variant and harboring the specific mutations (B.1.1.529 BA.1 mutated pool, Miltenyi Biotech, Bergish Gladbach, Germany). As a control, the same peptide pools, but with the original (Wuhan strain) sequence, were used (reference pool, Miltenyi Biotech, Bergish Gladbach, Germany). Following 2 h of incubation at 37 °C, 5% CO_2_, brefeldin A (5 μg/mL) was added, followed by an additional 4-h incubation at 37 °C, 5% CO_2_. Finally, the cells were fixed with 15 min incubation in PBS 2% PFA at room temperature and stained (as described above) using the fluorochrome-conjugated antibodies listed in Appendix A. The samples were acquired on a BD LSR II flow cytometer (BD Biosciences, Franklin Lakes, NJ, USA) with FACSDiva Software and analyzed by FlowJo v10 Software. The gating strategy used is presented in Appendix A.

Evaluation of the SARS-CoV-2 spike-specific B cells: For the B cells evaluation, 2 million PBMCs were stained for 30 min at 4 °C with fluorochrome-conjugated antibodies listed in Appendix A, then washed with a PBS/EDTA buffer, and incubated for 5 min with 7-AAD, for the viability evaluation. The recombinant biotinylated SARS-CoV2 Wuhan and Omicron spike proteins (Miltenyi Biotech, Bergish Gladbach, Germany) were conjugated separately with streptavidin PE, PE-Vio770, APC and Vioblue, for 15 min at room temperature, together with the fluorochrome-conjugated antibodies listed in Appendix A, before being added to final staining mix. The cells were then incubated with the final mix for 30 min at 4 °C. The samples were acquired on a BD LSR II flow cytometer (BD Biosciences, Franklin Lakes, NJ, USA) with FACSDiva Software and analyzed by FlowJo version 10 Software (BD Biosciences, Franklin Lakes, NJ, USA). The gating strategy used is presented in Appendix A.

Statistics: The paired, non-parametric Wilcoxon signed rank test was used to compare each subjects’ timepoints with one another. Mann–Whitney’s U test was used to compare each subjects’ memory from the Wuhan-Omicron B cell frequency to the Wuhan-only specific memory B cells. The *p* values equal to or less than 0.05 were considered statistically significant.

## 3. Results

### 3.1. Characteristics of PLWH Receiving a Fourth mRNA Vaccine Dose

All of the participants (*n* = 8) were people living with HIV (PLWH), who received the fourth dose of the original BNT162b2 mRNA vaccine in March 2022. The main demographic and clinical features are summarized in Table 1. The majority of the participants presented very low to non-detectable HIV blood viremia. Three subjects had developed AIDS and AIDS-associated comorbidities in the disease course and five individuals out of eight presented comorbidities unrelated to the HIV infection (Table 1). All patients underwent or were receiving antiretroviral therapy (ART) (Table 1) and received the third dose within 3 to 5.5 months, before being administered the fourth dose (Table 2). Notably, six subjects out of eight presented with CD4+ T cells counts below 500 cells/mm^3^, and a subsequent lower CD4/CD8 ratio, than average healthy subjects. These measurements are consistent with the CD4/CD8 ratio resulting from the lymphocytes’ subset phenotyping of PBMCs isolated at the pre-fourth dose timepoint (Table 2), that is, on average, lower (0.8) than the mean ratio found in healthy individuals (>1). All of the participants were negative for anti-nucleoprotein IgGs before the study’s enrolment, at the basal time (Table 2). In two individuals (V7 and V8), at the 2 months evaluation, anti-N IgG were detected, indicating the concomitant exposure to SARS-CoV-2 during the last month of the study period. In these two cases, the infection occurred without symptoms. None of them was tested with a nasopharyngeal swab.

### 3.2. SARS-CoV-2 Humoral Response after the Fourth Vaccine Dose Is Detectable in PLWH

Sera were collected for the spike-specific antibody measurements. All participants showed titers of anti-spike IgMs below the cut-off at all timepoints (Figure 1A). The IgGs targeting spike (Figure 1B) presented an increasing trend from the basal timepoint to 1 month after the fourth injection, and a significant decrease 2 months after the vaccine administration. All PLWH presented with anti-spike IgG titers above the cut-off value at all timepoints. Interestingly, for patient V8 (blue line) the anti-S IgGs peaked 7 days after the administration of the fourth dose, and then decreased similarly to other cases (Figure 1B). All subjects showed a higher concentration of the neutralizing antibody levels than the cut-off value at all timepoints (Figure 1C). The levels of IgGs targeting the receptor binding domain (RBD) of the Wuhan spike (Figure 1D), presented a significant increase in all subjects from the pre-vaccine timepoint to 7 days, followed by a decline after 1 month. At 2 months after receiving the fourth dose, the majority of patients showed an evident decrease, while two out of eight (V6 and V7) presented an increment of anti-RBD IgGs and V8 exhibited quite stable levels of these IgGs (Figure 1D). Participants V7 and V8 were both positive for the detection of IgGs targeting SARS-CoV-2 nucleoprotein 2 months post-injection, whereas in V6, the titers were below the cut-off value (Table 2). Notably, the levels of anti-RBD IgG and IgG targeting the total spike resulted above the cut-off value at all timepoints for all patients enrolled.

### 3.3. Fourth Vaccine Dose Induces the Transient Reactivation of the Wuhan Spike-Specific T and B Cells from Day 7 Post-Administration

Given the particular condition of HIV+ patients concerning CD4+ T cells, we focused on this subset. In particular, we investigated the ability of the CD4+ T cells to be re-activated in vitro (CD154+ cells) and produce the major effector cytokines (Figure 2A), i.e., IL-2, TNF-α and IFN-γ, upon stimulation with a pool of peptides covering the entire sequence of the Wuhan strain (wild type) spike, whose mRNA is used in mRNA-1273 and BNT162b2 vaccine formulations. The longitudinal trends for each patient displayed an average increase of the frequency of the spike-specific T cells in a time included between 7 days and 1 month after the booster dose. This was followed by a decline 2 months after the injection (Figure 2B). A similar pattern was observed for the spike-specific B cells, that were identified as CD19+ cells binding the recombinant spike protein, labelled in both PE and PE-Vio770 fluorochromes (Figure 2C). We observed that the frequencies of the spike-specific B cells increased variably between the study participants in the first month after the booster, and then was significantly lowered at 2 months after injection (Figure 2D).

### 3.4. Identification of the CD4+ T and B Cells Binding to the B 1.1.529 SARS-CoV-2 Omicron-Spike

Two months after the fourth vaccine dose, the PBMCs of the PLWH were stimulated separately with a peptide pool covering the entire spike of the Wuhan strain of SARS-CoV-2 (wild type, WT), a peptide pool covering the Wuhan spike portion, which is mutated in the B.1.1.529 (Omicron) variant of concern, and a peptide pool covering the mutated sequence (Figure 3A). The cells were selected for activation marker CD154 expression and for the production of at least one within the following effector cytokines: IL-2, TNF-α and IFN-γ. Figure 3B shows that the frequency of the CD4+ T cells reactive to the mutated region (both WT and Omicron sequences) of the spike protein was significantly lower than the frequency of those reactivated by the total Wuhan spike protein, and represented a minor portion of the spike-specific response in all subjects analysed. As for the comparison between the T cells, specifically reactive to the mutated sequence and its matching region in the Wild Type strain, no significant differences were observed, but only a slightly higher frequency of the WT-specific T cells producing at least one effector cytokine. Moving to the B cells, we focused on the memory (CD27+) B cells, being long-lived B cells generated during the primary immune responses to the T-dependent antigens. Re-exposures to the antigens induce a rapid reactivation, which results in the generation of antibody-producing plasma cells. To evaluate the possibility of the spike-Specific B cells’ cross-reactivity for recent VOCs, we identified the portion of the memory B cells able to bind the Omicron spike within the WT spike-specific subset (Figure 3C). The frequency of the CD27+ Wuhan spike-specific B cells, evaluated in a total of five patients, able to bind both the WT and the Omicron recombinant spike is significantly higher than the average percentage of the CD27+ B cells that exclusively bind the WT spike (Figure 3D). We performed the same analysis, inverting the gating strategy (Appendix A), to identify the frequency of the B cells binding only the variant recombinant spike and found they present a very low mean frequency (0.06, SD = 0.02). 

## 4. Discussion

As the SARS-CoV-2 vaccination campaign started to roll out between December 2020 and January 2021, several studies on the effectiveness of the vaccine were carried out on different cohorts all throughout 2021 [29,30]. The SARS-CoV-2 associated disease (COVID-19) can be developed in asymptomatic or mild forms, but may also cause severe dysregulation, leading to hospitalization and a poor outcome [31]. PLWH are a category of individuals whose immune dysfunction places them at risk of developing severe COVID-19 [22], and they represented a very small portion of the cohorts enrolled for the vaccines’ trials [13]. There are only a few studies reporting data on the vaccine-induced protection in this population [23,32,33,34]. As a matter of fact, the urgency now is to understand to what extent these individuals are protected and what booster strategy needs to be adopted, to guarantee the effectiveness of the vaccinations. To this end, we performed a longitudinal prospective study to evaluate the immunological reactivity to the fourth vaccine dose on a small cohort of PLWH.

Patients enrolled were all diagnosed with the HIV infection, were or are currently administered with ART and the majority of them presented a lower CD4/CD8 ratio than the average population. They received the fourth dose between 3 and 5.5 months from the previous vaccine dose. The vaccine-induced immunization was confirmed by the negativity of IgGs targeting the SARS-CoV-2 nucleoprotein at the pre-fourth injection time. Past studies analyzing larger at-risk cohorts following the two doses of the COVID-19 vaccination cycle, demonstrated that IgG levels can be compared to healthy individuals’ [35]. In the present cohort, we observed that the levels of anti-spike IgGs stay highly above the cut-off value throughout all of the timepoints, as well as the levels of the neutralizing antibodies. This suggests that the humoral protection persists over time in PLWH, at least to the conserved portions of the protein, which represent the majority of the spike epitopes. Of note, despite an initial increase, anti-RBD IgGs levels showed a decrease 2 months after the fourth dose for five cases out of eight and were quite stable in 1 individual of this cohort. Indeed, one of the patients showing an increase in the levels of anti-RBD IgGs and the patient with high levels at all timepoints were found positive for anti-nucleoprotein IgGs, confirming they were likely infected by SARS-CoV-2—possibly one of the most recent variant strains (Omicron)—in a moment between 1 month and 2 months following the fourth vaccine dose. Intriguingly, the titers of IgGs targeting the total spike in these patients decreased, following the same trend as the other participants’. Nevertheless, focusing on the absolute titers of anti-S IgGs, they are higher than anti-RBD IgGs’ and include all of the antibodies targeting the entire spike sequence; thus, anti-RBD antibodies might be diluted in a wider pool of the spike-specific IgGs. Of note, these individuals did not recall any signs or symptoms of infection, enforcing the concept of the effectiveness of the booster dose, in terms of the prevention from severe COVID-19. 

Focusing on the T cell response, no specific impairment in their induction or functionality was found. We observed an immediate reactivation 7 days after the second booster, followed by a general decrease of the effector cytokine production at 2 months after the injection. The mean trend of these eight PLWH showed spike-specific CD4+ T cell frequencies that are comparable to vaccinated healthy individuals’ [29]; taking into account that the latest (up to three months before basal time) absolute count of the CD4+ T cells was below the normal range in many cases (V3, V4, V5, V6, V7), the transient reactivation of the spike-specific T cells is efficient even in PLWH with lower CD4+ T cell counts and results in the effector cytokine production, in most cases. Moving to the B cells, consistently with the spike-specific T cell response, we found a similar trend of increase in the spike-specific B cell detection until 1 month after the fourth vaccine injection and a significant decline of the spike-specific B cell frequency at the 2 month timepoint. Considering we analyzed the circulating PBMCs, a possible reason for this trend, especially in those two individuals positive for anti-N IgGs at the 2 month timepoint, is that the CD19+ B cells responding to the last vaccine dose, or infection, are detectable in the bloodstream up to 1 month post-injection, before decreasing and likely migrating to the secondary lymphoid organs [36]. 

As a consequence of the rapid advent of the new variants of concern (VOCs), new waves of the pandemic have been experienced worldwide, despite an overall better outcome and disease course in vaccinated individuals [37,38]. This suggests that, while the vaccination-induced immunity may be not that efficient in avoiding or promptly resolving the infection by VOCs, it protects from severe COVID-19. We tested the responsiveness of T cells to those spike epitopes that are found mutated in the B.1.1.529 (Omicron) strain at 2 months after the fourth vaccine dose. We found that the CD4+ T cells are majorly reactive to the conserved portions of the protein, showing minimal activation when stimulated with the mutated antigens alone. This result confirms that the greater part of the spike-specific T cells is activated by the conserved antigens within the variants, as reported previously [39], and a minor portion presents the ability of the mutated antigen recognition; this suggests the scarce frequency of the cross reactive T-cells, meaning the T cells that are activated by the antigens carrying mutations, with respect to the antigens for which the TCR is originally specific, possibly suggests a low-frequency cross-reactivity. As for the assessment of the B cells’ memory, we performed a staining procedure to identify the memory B cells capable of binding both the Omicron (B.1.1.529) and Wuhan (WT) spikes, by incubating both the entire recombinant spike proteins with the participants’ circulating B cells. We found that, in all patients analysed, a high frequency of the Wuhan binding memory (CD27+) B cells can also bind the Omicron spike and a lower percentage only binds to the Wuhan spike. Despite that the memory B cells were not tested against the isolated mutated sequences of the spike, our observations suggest that the vaccine-induced B cells might recognize both the conserved and possibly mutated sites in the Omicron spike protein as well. These observations are in line with the previously published data [40]. Moreover, past publications [41] speculate that the cross-reactivity could be due to the incidence of infections by low-pathogenetic coronaviruses preceding the outbreak of the SARS-CoV-2 pandemic. Still, we did not investigate the functionality of the Wuhan-Omicron spike specific B cells upon the binding and activation. In addition, it could be argued that the increasing number of infections by the Omicron variant is due to a less efficient, or absent, blocking activity against the mutated RBD sequences mediated by the neutralizing antibodies secreted after the vaccine administration [42]. 

Our study presents some limitations; first of all, a greater number of subjects is needed to draw more solid conclusions about the efficiency of the vaccine-induced immune response to SARS-CoV-2, in PLWH and patients with AIDS. Furthermore, we did not test the neutralization activity of the vaccination-induced antibodies against the latest VOCs, which would yield a wider insight into the effective humoral protection given by the vaccine, in response to the infection by insurgent VOCs. Moreover, while the T cells were tested using specific Omicron-mutated antigens, the B cells’ binding ability was assessed using the entire Omicron spike protein, meaning we cannot distinguish whether the B cells specifically bind the mutated RBD or conserved regions of the spike. This could be very interesting to investigate, to prove the presence of the cross-reactive B cells, i.e., specific for the Wuhan spike, but also able to bind the shared and mutated sites of Omicron’s. 

## 5. Conclusions

In conclusion, we observed a transient reactivation of the spike-specific immunity in this small cohort of PLWH, in response to the fourth dose of the original SARS-CoV-2 mRNA vaccine. The spike-specific cellular response showed an average increasing trend up to 1 month after the fourth injection, and then declined at the 2 month timepoint. Similar kinetics is presented by the humoral response even if the spike-specific IgG levels are over the cut-off values at all time points. This supports that the fourth vaccine dose transiently boosts a pre-existing immunological memory in PLWH. Our data also open an insight into the evaluation of the cross-reactivity of the vaccine-induced response against the SARS-CoV-2 variants of concern. Despite many limitations, this work adds useful and unprecedented information to a growing body of data on the humoral and cellular immune response induced by the second booster dose, in a small cohort of at-high-risk individuals, represented by PLWH with lower CD4+ T cell counts.

## Figures and Tables

**Figure 1 biomedicines-10-03261-f001:**
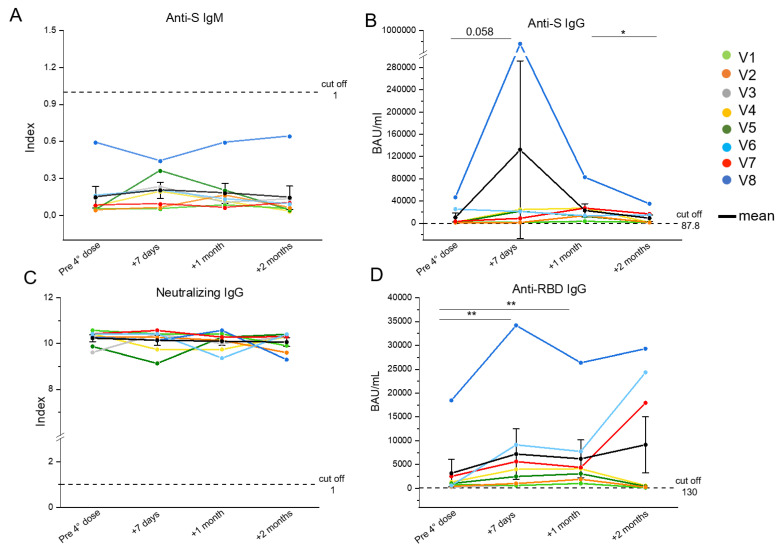
**Evaluation of the SARS-CoV-2-specific humoral response in eight people living with HIV (PLWH) up to 2 months after the fourth vaccine dose**. Anti-S IgM (cut off = 1) (**A**), anti-S IgG (cut off = 87.8) (**B**), neutralizing IgG (cut off = 1) (**C**), and anti-RBD IgG (cut off = 130) (**D**) serum levels were evaluated in PLWH before, one week, one month and two months after the fourth vaccine dose. Each color corresponds to a different subject. Black dots indicate mean values. * *p* ≤ 0.05, ** *p* ≤ 0.005. Error bars indicate SE values.

**Figure 2 biomedicines-10-03261-f002:**
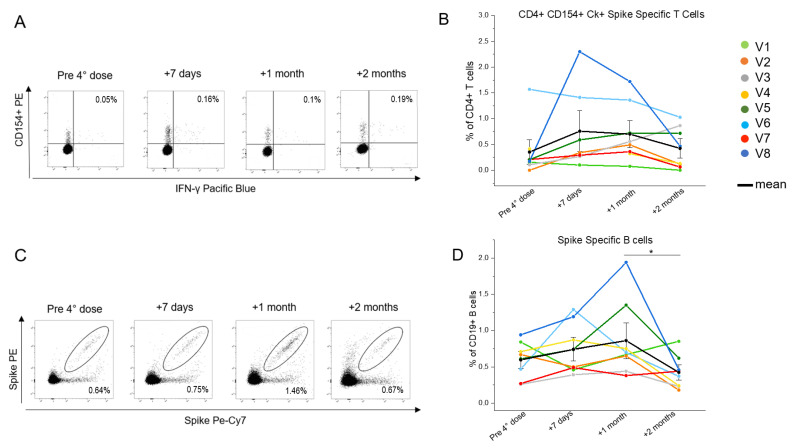
**Evaluation of the SARS-CoV-2-specific circulating CD4+ T and B cells in eight people living with HIV (PLWH), up to 2 months after the fourth vaccine dose**. (**A**) Representative flow cytometric plots of the spike-specific CD154+CD4+IFN-γ+ T cells in one selected individual living with HIV. (**B**) Kinetic analysis of the frequencies of the CD154+CD4+ T cells producing at least one cytokine among IL-2, IFN-γ, and TNF-α in eight PLWH before, one week, one month, and two months after the fourth vaccine dose. Each color corresponds to a different subject. The T cell evaluation of subject V4 lacks the +7 days timepoint. Black dots indicate mean values. (**C**) Representative flow cytometric plots of the spike-specific B cells in one selected individual living with HIV. (**D**) Kinetic analysis of the frequencies of the spike-specific B cells in eight PLWH before, one week, one month, and two months after the fourth vaccine dose. Each color corresponds to a different subject. Black dots indicate mean values * *p* ≤ 0.05. Error bars indicate SE values.

**Figure 3 biomedicines-10-03261-f003:**
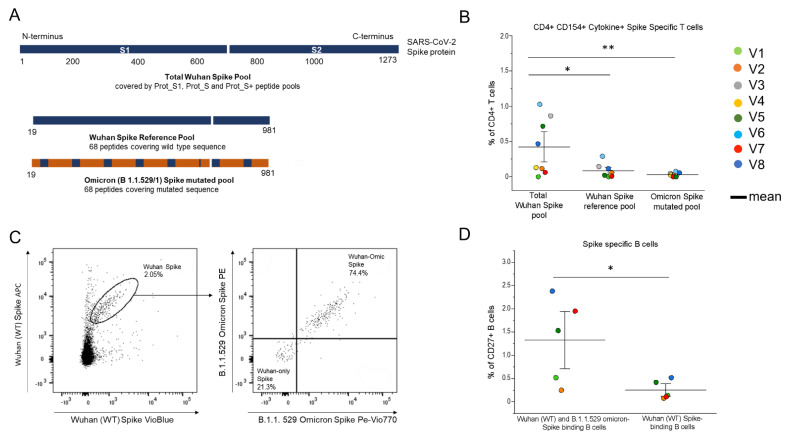
**Assessment of the CD4+ T and B cells responses specific to the Wuhan SARS-CoV-2 spike and the B 1.1.529 SARS-CoV-2 VOC spike, 2 months after the administration of the fourth vaccine dose in eight vaccinated PLWH vaccinated**. (**A**) Visualization of the SARS-CoV-2 spike protein (1273 aa) covered by the S1, S+ and S peptide pools, the wild type sequence and the mutated sequence (aa 19–981) in the Omicron B1.1. 529 (BA.1). All peptide pools contain 15-mer peptides, 11 aa overlapping. (**B**) The frequencies of the CD154+ cells producing at least 1 cytokine among IL-2, IFN-γ and TNF-α, stimulated with the spike peptide pool, wild type (Wuhan) or the B. 1.1.529 (Omicron) peptide pool covering the mutated sequence of the spike. (**C**) Representative plots of the identification of the Wuhan spike-specific, labelled with APC-PB, and the Omicron spike-specific B cells, labelled with PE-PeVio770 within the Wuhan spike-specific pool. (**D**) Frequencies of the spike specific memory B cells binding both the Wuhan spike and Omicron spike and the B cells binding exclusively to the Wuhan spike. Each color corresponds to a different subject. Black lines indicate mean values.* *p* ≤ 0.05, ** *p* ≤ 0.005. Error bars indicate SE values.

**Table 1 biomedicines-10-03261-t001:** Demographic and clinical features of eight people living with HIV (PLWH), who received the fourth anti-SARS-CoV-2 mRNA vaccine dose. V: vaccinee; AIDS: acquired immunodeficiency syndrome; AML: acute myeloid leukemia; C: original BNT162b2 vaccine; ART: antiretroviral therapy; HSCT: hematopoietic stem cell transplantation; MAC: mycobacterium avium complex; NA not applicable; ND not determined; S: original mRNA-1273 vaccine; SD: standard deviation; TND: target not detected.

PLWH	Age at Fourth Dose Vaccination (Years of Age)	Age at HIV Diagnosis (Years of Age)	Time from HIV Diagnosis to ART Start (Days)	CD4+ T Cell Nadir Count (Cell/mm^3^)	HIV-1 RNA Zenith (Copies/mL)	AIDS (Years of Age)	AIDS Event	Comorbidities	Latest CD4+ T Cell Count (Cell/mm^3^)	Latest HIV-1 RNA (Copies/mL)	Adherent to ART	Vaccine Type (First + Second + Third + Fourth Dose)
V1	47	22	29	355	>500,000	NA	No	Allogenic HSCT in AML at 41 years	1300	TND	Yes	S + S + C + C
V2	64	31	1034	300	ND	NA	No	Cardiovascular disease	989	TND	Yes	S + S + C + C
V3	64	50	42	8	227,015	Yes (50)	Cryptococcal meningitis and pneumonia; MAC infection	Major depressive disorder	172	<20	No	S + S + S + C
V4	65	30	1461	15	231,833	Yes (60)	*Pneumocystis jiroveci* pneumonia; CMV infection	Chronic hepatitis C	253	22,000	No	S + S + C + C
V5	48	45	3	3	1,420,000	NA	No	None	210	<20	Yes	S + S + S + C
V6	58	48	28	28	>10,000,000	Yes (55)	Wasting syndrome; esophageal candidiasis	None	243	58	Yes	S + S + S + C
V7	51	31	128	103	449,000	NA	No	Severe obesity	390	TND	Yes	S + S + C + C
V8	71	70	7	186	2,810,000	NA	No	None	829	84	Yes	S + S + S + C
Mean	58.5	40.9	341.5	124.7	-	-	-	-	548.2	-	-	-
SD	8.9	15.4	572	140.3	-	-	-	-	430.8	-	-	-

**Table 2 biomedicines-10-03261-t002:** **Cellular ratio, measurement of anti-nucleoprotein IgG levels and time from the third dose administration of eight people living with HIV (PLWH), who received a fourth anti-SARS-CoV-2 mRNA vaccine dose**. SD: standard deviation.

PLWH	CD4/CD8 Ratio	Anti-N IgG * T0 Pre-Fourth Dose	Anti-N IgG * T +7 Days	Anti-N IgG * T +1 Month	Anti-N IgG * T +2 Months	Time between the Third and Fourth Vaccine Doses (Days)
**V1**	1.7	0.01	0.01	0.01	0.01	167
**V2**	1	0.02	0.02	0.02	0.03	160
**V3**	0.4	0.08	0.06	0.02	0.02	107
**V4**	1	0.01	0.06	0.02	0.01	108
**V5**	0.4	0.08	0.08	0.08	0.07	101
**V6**	0.8	0.56	0.56	0.50	0.56	94
**V7**	0.1	0.04	0.03	0.03	3.38	117
**V8**	0.7	0.92	0.64	0.41	8.14	94
Mean SD	0.8 0.5	0.2 0.3	0.2 0.3	0.1 0.2	1.5 2.9	119 29

***** index, cut-off = 1.4.

## Data Availability

Data is available upon reasonable request.

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
