# Peer review of "Fourth Dose of mRNA COVID-19 Vaccine Transiently Reactivates Spike-Specific Immunological Memory in People Living with HIV (PLWH)"

_biomedicines, 2022, doi:10.3390/biomedicines10123261_

Round 1

Reviewer 1 Report

In the article 'Humoral and cellular immune response reactivation induced 2 by SARS-CoV-2 fourth vaccine dose in people living with HIV (PLWH)' a study of immune response to the fourth vaccine dose in people living with HIV involving small group of participants is presented. The main conclusion is that the fourth vaccine induces immune response. It is clear that the investigation of such specific groups of individuals are very needed. In my mind, the current research presented in this article has several drawbacks which are addressed in the comments below.

1. The title has no message in it.

2. line 33 delete "most"

3. lines 52-58. This information is additional and not directly related to the article.

4. Please indicate in methods in lines 101; 109; 134 how many replicates of the experiments were performed, please add the error bars to the results presented in figures.

5. Lines 216-217. It is very difficult to understand what kind of peptide pools were used even though they are specified in the Methods. I suggest to use somekind of abbrevations or explain the rationale of peptide pools choice in methods.

6. line 222. At least to grammar mistakes.

7. line 225-226. Nowhere in the article there is explained why the authors consider that the release of at least 1 cytokine is significant.

8. Figure 1. Please change the color of 'V6' which is almost the same as 'mean'.

9. line 282. Please use CD4/CD8 ratio.

10. line 284. change to "negativity of".

11. line 293-295. Which exact cases you think had COVID-19 between months 1 and 2? Please elaborate why there is no increase of anti-S IgG? Conversely, anti-RBD IgG increased... A test of anti-N IgG could clarify the speculation here.

12. Statment in lines 310-313 should be backed up by appropriate reference.

13. lines 324-325. Please explain what is meant her? Low cross-reactivity? Could it be related to the aminoacid sequence differences/similarities?

14. line 327. delete "the"

15. lines 358-359. In the article now sufficient evidence is presented for this conclusion.

Reviewer 2 Report

Dear Editor,

thank you for the opportunity to review the manuscript entitled Humoral and cellular immune response reactivation induced 2 by SARS-CoV-2 fourth vaccine dose in people living with HIV 3 (PLWH).

Manuscript is well written. The English is well use and it is easily to go through the sections. It is carefully designed and presents the effects of a second booster on the reactivation of Spike-specific humoral and cell-mediated immune responses in prioritized population for SARS-CoV-2 fourth vaccine dose. Only 8 participants were enrolled due to the specific inclusion criteria. All had no signs of previous SARS-CoV-2 infection. Results were presented in details. This study has limitations and authors mentioned them in discussion. Although, the study has limited number of participants and therefore, it is difficult to bring solid conclusion, in my opinion the findings are valuable. I do not have further comments for authors.

Best regards!

Reviewer 3 Report

The manuscript investigates the humoral and cellular immune response reactivation induced  by SARS-CoV-2 fourth vaccine dose in a small number of  people living with HIV. The data is clearly presented and provides strong evidence for encouraging uptake of booster vaccines in those with immunosuppression. They also demonstrate some cross reactivity in terms of previous vaccines that may contribute towards the improved outcomes for patients that become re-infected with SARS-CoV-2.

There are some points that need addressing.

1) Line 164 - 'for ant-Nucleoprotein IgGs' - should read anti-Nucleoprotein.

2) Line 283 - 5,5 months - should read 5.5 months

3) Line 316 - reference '342' - should read reference 34

4) Supplementary Table 1. Please check the fluorochromes for CD3 V450 and CD45 V450-C. Should the latter be V500?

5) Supplementary figure 1 - Please explain why CD16 was used for identifying NK cells? This would exclude CD56bright positive NK cells that are typically CD16-ve.

6) In the supplementary figures, populations such as NK cells and NKT cells are demonstrated/identified but not refered to in the main text of the study. Please provide the rationale for this.

Reviewer 4 Report

Lamacchia G et al. described the humoral and cellular response after the fourth dose of SARS-CoV-2 mRNA vaccines in people living with HIV. The authors observed a detectable spike-specific IgG response and T cell response although this the response declined 2 months after the administration of the vaccine. Moreover, they also described that both spike-specific CD4+ T and B cells were reactive to conserved regions of spike protein, being capable of binding both Wuhan and Omicron-spike entire protein.

The study is well designed and explained, and the results of the immune response after the fourth dose of SARS-CoV-2 vaccine against different VOCs in HIV infected subjects have not been described yet and might be relevant findings for the improvement of vaccination strategies in this susceptible population. However, one important limitation is the sample size of the study, which has been already indicated by the authors in the limitation section, and the differences in clinical characteristics of the patients might be a problem for a clear interpretation of the results, mainly in the data with high dispersion. Additionally, a more complete follow-up of the subjects might be suitable for a better understanding of the evolution of the vaccine response of each patient, including data after the first, second or third doses of the vaccines, or even a longer period after the fourth dose, as possible.

Major comments:

-       Regarding the design of the study, I would recommend to include the data of the immune response in these patients after the first, second or third dose of SARS-CoV-2 vaccines, I think that is an important information for a better understanding of the evolution of the subjects.  

-          Data for all mentioned experiments should be showed. As it is described in the gating strategy (Supplementary Figure 1), NK cells and NKT cells were also analyzed by flow cytometry but these results are not mentioned or showed in the article. Moreover, the identification of NK cells is not correct, NK cells should be selected as CD56+ (CD16 is used to differentiate NK subsets): CD56Bright (CD16dim/neg), CD56Dim (CD16+) and CD56Neg (CD16+)

-          Any test for COVID-19 was done before the study? In line 294 the authors hypothesized that 2 of the subjects could have been infected at this time point. If no Covid-19 test was done, the authors should measure N-specific antibodies in plasma in order to discard previous infections.  

Minor comments:

-          I suggest to modify a little bit the introduction. The first paragraph is too basic and general, it is not required to explain HIV infection, its phases or AIDS symptoms. Instead of explaining this, I would include information about how is the immune response to the fourth dose in general population and other risk populations in the last paragraph.

-          Line 95: Specify when exactly was obtained the sample of basal time. How much time before the fourth dose? I assume that is after the third dose.

-          Lines 115 and 130: explain the protocol for the staining with fluorochrome-conjugated antibodies (temperature, time of incubation, washes etc).

-          Gating strategy (Supplementary Figure 1): did the authors use a viability marker in this case? A viability marker is required for all the measurements with frozen cells.

-          Table 1: There is no name and figure legend in this table.   

-          Figure 1D: regarding anti-RBD IgG response, two groups of patients can be distinguished 2 months after the vaccination, one with high response and the other with a very low response, very close to the cut-off. How the authors explain this? In spite of the low sample size, I suggest to compare the clinical characteristics of these two groups to clarify if this difference might be due to the disease progression or patient’s clinical conditions. The same is observed in Figure 3A (total Wuhan Spike pool), I recommend to do the same and to check if the patients that showed higher % of CD154+ cytokine+ CD4+ T cells are the same that displayed higher anti-RBD IgG levels. These data should be also discussed.

-          In the discussion, I would highlight that the fourth injection did not generate an effective immune response if we focused on 2 months after the vaccination comparing with the pre-vaccination time point, in most of the determinations (for example spike-specific cellular response, both T and B, or anti-S IgG production). For a suitable boosting, the fourth dose should induce a higher humoral and cellular response two months after. If there is not any improvement in the immune response comparing with the basal time point two months after the vaccination (it is not a long period of time), the author should not conclude that the fourth infection effectively boosts a pre-existing immunological memory (line 358), it is a very risky affirmation.

Round 2

Reviewer 1 Report

In the article 'Fourth dose of mRNA COVID-19 vaccine transiently reactivates Spike-specific immunological memory in people living with HIV (PLWH)' a study of immune response to the fourth vaccine dose in people living with HIV involving small group of participants is presented. It is clear that the investigation of such specific groups of individuals are very needed. The manuscript after the revision is suitable for publication after minor text editing such as a dot after the title, etc.

Reviewer 4 Report

Taking to account the modifications of the article, I suggest the article to be accepted in present form.